# Impact of a Pretreatment Step on the Acidogenic Fermentation of Spent Coffee Grounds

**DOI:** 10.3390/bioengineering9080362

**Published:** 2022-08-03

**Authors:** Joana Pereira, Marcelo M. R. de Melo, Carlos M. Silva, Paulo C. Lemos, Luísa S. Serafim

**Affiliations:** 1CICECO-Aveiro Institute of Materials, Departamento de Química, Universidade de Aveiro, 3810-193 Aveiro, Portugal; joanasofiapereira@ua.pt (J.P.); marcelo.melo@ua.pt (M.M.R.d.M.); carlos.manuel@ua.pt (C.M.S.); 2LAQV-REQUIMTE, Department of Chemistry, Faculty of Science and Technology, Universidade NOVA de Lisboa, 2829-516 Caparica, Portugal; paulo.lemos@fct.unl.pt

**Keywords:** acidogenic fermentation, spent coffee grounds, short-chain organic acids, pretreatments, hydrolysis

## Abstract

Acidogenic fermentation (AF) is often applied to wastes to produce short-chain organic acids (SCOAs)—molecules with applications in many industries. Spent coffee grounds (SCGs) are a residue from the coffee industry that is rich in carbohydrates, having the potential to be valorized by this process. However, given the recalcitrant nature of this waste, the addition of a pretreatment step can significantly improve AF. In this work, several pretreatment strategies were applied to SCGs (acidic hydrolysis, basic hydrolysis, hydrothermal, microwave, ultrasounds, and supercritical CO_2_ extraction), evaluated in terms of sugar and inhibitors release, and used in AF. Despite the low yields of sugar extracted, almost all pretreatments increased SCOAs production. Milder extraction conditions also resulted in lower concentrations of inhibitory compounds and, consequently, in a higher concentration of SCOAs. The best results were obtained with acidic hydrolysis of 5%, leading to a production of 1.33 gSCOAs/L, an increase of 185% compared with untreated SCGs.

## 1. Introduction

Wastes are no longer seen as residues but as profitable opportunities and a potential feedstock for bioprocesses within the biorefinery concept [1]. Food wastes stand out, not only due to their low price, high production, and availability, but also because their complex composition allows for the production of different final products [2].

Coffee is the second most consumed beverage in the world, beaten only by water [3]. The industry has seen an increase in production of 77% in the last 30 years, with a global production of 9.9 megatons of coffee beans in 2019/2020 [4]. However, a substantial amount of residue formation, particularly spent coffee grounds (SCGs), is associated with this high production rate. SCGs are the residue from the brewing and the production of soluble coffee processes and are estimated to be in a ratio of 650 kg per ton of green coffee beans processed [3,5]. Due to their high production and availability, SCGs have been used as a substrate for the production of several bioproducts such as biofuels and biomaterials, and the possibility of an SCG-based biorefinery has been proposed by several authors [3,6,7,8].

Among products that can be obtained from industrial residues are short-chain organic acids (SCOAs)—aliphatic monocarboxylic acids composed of six or fewer carbon atoms such as acetic, propionic, butyric, valeric, and lactic acids [9]. These metabolites are usually based on petrochemical resources and have a wide range of applications in several industries [10]. However, the environmental impact associated with those processes and increasing crude oil prices have resulted in a growing interest in biological production, especially using industrial organic residues [11]. SCOAs produced using this strategy are particularly interesting for the biosynthesis of polyhydroxyalkanoates (PHA), where the manipulation of the mixture of acid composition was found to play an important role in the biopolymer molecular composition and, consequently, their thermochemical characteristics [12].

SCOAs are intermediary metabolites of anaerobic digestion, produced during the acidogenic and the acetogenesis steps of anaerobic digestion. Their biological production usually consists of avoiding the conversion of SCOAs into methane by inhibiting the methanogenesis step through the manipulation of the operating conditions to hinder the survival of methanogenic bacteria [13]. Although only a few authors have reported the use of SCGs to produce SCOAs via acidogenic fermentation [14,15], its use for methane production is well-documented [16,17,18,19,20], which shows the potential of this strategy. Despite the final product, anaerobic digestion starts with the hydrolysis step, which is usually the limiting step, particularly when working with complex wastes [10]. Since most of the feedstocks used in acidogenic fermentation (AF) are recalcitrant and are not readily biodegradable, the inclusion of a pretreatment step is often required [21]. Due to the lignocellulosic nature of SCGs, a pretreatment step could increase their biodegradability by fragmenting lignin and carbohydrates into simpler molecules—a strategy that would increase the overall productivity of the process.

Several pretreatment methods of SCGs have already tested and reported in the literature to unlock the sugar potential of this residue or extract interesting compounds, such as lipids, oils, or antioxidants [3,22,23,24]. The approaches reported for sugar recovery often resort to hydrolytic processes, with acidic hydrolysis as the most used pretreatment strategy [22,23,25,26]. Nonetheless, pretreatment methods such as basic hydrolysis [17,27,28], enzymatic hydrolysis [29,30], hydrothermal pretreatment [31,32], microwaves [33,34,35], ultrasounds [36,37,38], subcritical water [24,39,40], and supercritical CO_2_ extraction [24,34,41] have already been tested by several authors. After a careful revision of the literature, a set of pretreatments for SCGs were chosen to be tested in the present study: acidic hydrolysis with 5 and 10% of H_2_SO_4_, basic hydrolysis with 5% NaOH, hydrothermal pretreatment, microwave, ultrasounds, and supercritical CO_2_.

The breaking down of the SCG structure releases not only sugars but also other molecules such as furfural and 5-hydroxymethylfurfural (HMF) from the degradation of cellulose and hemicellulose, phenolic compounds mainly resulting from the degradation of lignin and other aromatic compounds, or Millard reaction products (MRPs) formed from sugar degradation [40,42,43]. These are often reported as having antimicrobial activity and can hinder biological processes. Furans and phenolic compounds can negatively affect microbial fermentation by inhibiting cell growth and sugar uptake rate since they are reported to affect metabolisms, cell wall formation, and DNA, plasmid, RNA, and/or protein synthesis [43].

The impact of inhibitors resulting from the pretreatment step of SCGs on biological processes was already observed by some authors. Kim et al. [20] reported that, while a high NaOH concentration and high temperature conditions improved the efficiency of the pretreatment of SCGs, they impacted negatively the anaerobic digestion process, decreasing the methane production. Obruca et al. [8] tested detoxification methods that reduced the polyphenols content in hydrolyzed SCGs by about 22%. This change led to an increase in the yield of polyhydroxyalkanoates by more than 25%. On the other hand, Hudeckova et al. [44] tested different dilutions of hydrolyzed SCGs for lactic acid production by *Lactobacillus rhamnosus*, and the best results were achieved with the undiluted hydrolysate. This might indicate a higher tolerance by the strain used or an insufficient amount of carbon provided [44]. These findings emphasize the need for a balance between the amount of sugars released and inhibitor product formation when selecting the pretreatment conditions.

In this work, to increase the amount of SCOAs obtained during the acidification of SCGs, several pretreatments were tested. Not only the prevalence of sugars and inhibitors were determined, but all the pretreated SCGs were used in AF assays. By employing this strategy, it was possible to understand the real impact of the pretreatment on the substrate uptake and, consequently, SCOAs’ production.

## 2. Materials and Methods

### 2.1. Substrate

SCGs were provided by the coffee shop at the Department of Chemistry of the University of Aveiro. The substrate was dried in an oven at 105 °C to constant weight for 24 h and stored in a desiccator.

### 2.2. SCG Pretreatments

Several pretreatment conditions were selected and performed with 6 g of SCGs dispersed in 160 mL of deionized water. For hydrothermal pretreatment, the SCG suspension was autoclaved for 1 h at 121 °C. Acidic hydrolysis was tested for two H_2_SO_4_ concentrations, 5% and 10%, for 1 h in the autoclave at 121 °C. Basic hydrolysis with 5% of NaOH was also conducted at 121 °C for 1 h in the autoclave. The ultrasonication pretreatment was performed at 40 °C and 400 W for 30 min (Elma Schmidbauer GmbH, Singen, Germany). SCG supercritical CO_2_ extraction was conducted at 300 bar and 50 °C, with a flow rate of 12 gCO_2_/min for 2 h (0.5 L Lab unit, model Speed-SFE from Applied Separations, Inc., Allentown, PA, USA), as reported by Melo et al. [34]. Finally, the SCGs were submitted to microwave at 800 W for 4 min. The outcome of each pretreatment was fully used on the following acidogenic assays. Whenever possible, 5 mL of the product of the pretreatment was filtered with a glass microfiber filter 629 with a pore diameter of 1 μm (VWR, Radnor, PA, USA) for the determination of monomeric sugars, COD, total phenolic compounds (TPC), and MRPs. All the assays were performed in single mode.

### 2.3. Acidogenic Fermentation Assays

#### 2.3.1. Inoculum

The mixed microbial culture (MMC) used as inoculum in this study was collected from an aerobic tank of the wastewater treatment plant Aveiro Sul, SIMRia (Aveiro, Portugal), and stored at 4 °C until utilization. The aerobic inoculum was used in order to avoid the presence of methanogenic bacteria and enhance acidification [13].

#### 2.3.2. Experimental Set-Up

Batch tests were conducted in encapsulated flasks with 100 mL of working volume, maintained at 28 °C with magnetic stirring at 300 rpm, and purged with N_2_ prior to incubation to ensure anaerobic conditions. SCGs submitted to the selected pretreatments were used as a carbon source in a proportion of 1 gCOD of SCG per 1 gCOD of MMC with mineral medium: 160 mg/L of NH_4_Cl, 160 mg/L of KH_2_PO_4_, 80 mg/L of CaCl_2_, 160 mg/L of MgSO_4_, 800 mg/L of NaHCO_3_, 200 mg/L of CoCl_2_, 30 mg/L of MnCl_2_, 10 mg/L of CuCl_2_, 100 mg/L of ZnSO_4_, 300 mg/L of H_3_BO_3_, 30 mg/L (NH_4_)_6_Mo_7_O_2_, and 20 mg/L of NiCl_2_. Nontreated SCGs were used as a control. Every day, a 2.0 mL sample was collected under anaerobic conditions and centrifuged at 13,000 rpm for 10 min (MiniSpin, Eppendorf, Hamburg, Germany). The pellet was discharged, and the supernatant was stored at −16 °C for further determination of SCOAs. All the assays were performed in single mode.

### 2.4. Analytical Methods

Due to the small volume of the assays and the high number of samples collected throughout the experiments, they were only analyzed once.

#### 2.4.1. Determination of SCOAs, Monomeric Sugars, and Furans

A total of 600 μL of each sample was filtered using Vecta Spin Tubes (Whatman, Piscataway, NJ, USA) with a membrane of 0.2 μm (Whatman, Kent, UK) at 8000 rpm (MiniSpin Eppendorf, Hamburg, Germany) for 20 min before HPLC injection. Monomeric sugar concentration released to the liquid phase was determined using a Rezex RPM-Monosaccharide Pb^+2^ (8%) column (Phenomenex, Torrance, CA, USA) at 85 °C and a refractive index detector (Merck, Darmstadt, Germany), using MilliQ water as eluent (0.6 mL/min). SCOAs concentration of the acidogenic tests was measured in a Rezex ROA-Organic Acid H^+^ (8%) column (Phenomenex, Torrance, CA, USA) at 65 °C and a refractive index detector (Merck, Darmstadt, Germany), using H_2_SO_4_ 0.005 N as eluent (0.5 mL/min). HMF and furfural were detected using the same column and conditions as the SCOAs, using a DAD detector (Merck, Darmstadt, Germany) instead. The calibration curves were performed frequently using freshly prepared standards in the range of 0–1 g/L for sugars and furans and 0–5 g/L for SCOAs to ensure the method linearity.

#### 2.4.2. Chemical Oxygen Demand (COD)

COD was measured with Spectroquant Kit (Merck Millipore, Darmstadt, Germany), and the solutions used were prepared according to Standard Methods [45]: a digestive aqueous solution with K_2_Cr_2_O_7_, HgSO_4_, and H_2_SO_4_ and an acid solution with H_2_SO_4_ and AgSO_4_. Next, a 2 mL of sample properly diluted was added to 1.2 mL of digestive solution and 2.8 mL of acid solution. The mixture was incubated at 150 °C for 2 h. After cooling, the absorbance at 600 nm was measured. The calibration was performed frequently using freshly prepared standards with glucose with COD concentrations between 0 and 1 g/L to ensure the method linearity.

#### 2.4.3. pH Measurement

The pH of the samples was measured using an electrode InPro 3030/200 (Mettler Toledo, Columbia, MD, USA) and a benchtop meter sensION^+^ MM340 (Hach, Loveland, CO, USA) at 25 °C. Prior to the measurements, the pH meter was calibrated using technical buffer solutions of pH 4.00 ± 0.02, 7.00 ± 0.02, and 10.0 ± 0.02.

#### 2.4.4. Browning Intensity

To have an insight into the formation of MRPs, the absorbance at 294 nm was used as an index of the uncolored intermediate compounds produced during the Maillard reaction, while the absorbance at 420 nm was used as an index of the brown polymers produced. A UV-Vis spectrophotometer (UVmini-1240, Shimadzu, Kyoto, Japan) was used, and the samples were diluted by 1:40.

#### 2.4.5. Total Phenolic Compounds (TPC)

TPC was measured using the Folin–Ciocalteau method, as described by Bravo et al. [46]. A volume of 500 µL of Folin–Ciocalteau reagent was mixed with 100 µL of sample and 7.9 mL of demineralized water. After a 2 min delay, 1.5 mL of a 7.5% (*w*/*v*) sodium carbonate solution was added and the samples were incubated in darkness at room temperature for 90 min. Afterwards, the absorbance at 765 nm was measured using a UV-Vis spectrophotometer (UVmini-1240, Shimadzu, Kyoto, Japan). A calibration curve was prepared frequently using freshly prepared standards of gallic acid, ranging from 0 to 1 g/L, to ensure the method linearity. The total phenols content was expressed in grams of gallic acid equivalent (GAE) per gram of SCG dry matter (mgGAE/g).

### 2.5. Calculations

#### 2.5.1. COD Conversions

For further calculations, the concentrations of SCOAs determined by HPLC were converted from g/L to gCOD/L using conversion factors that represent the mass (g) of oxygen required to oxidize 1 g of a compound based on the oxidation reactions for each compound. The overall oxidation equation is represented by:a compound + b O_2_ → c CO_2_ + d H_2_O + e NH_3_(1)
where a, b, c, d, and e represent the stoichiometric coefficients of the equation. Therefore, the conversion factor (cf) was calculated according to the following equation:(2)$$\mathrm{cf}\;\left({\mathrm{gO}}_{2}/g\right)=\frac{b\;\times \;\mathrm{MM}\left(O_{2}\right)}{a\;\times \;\mathrm{MM}\left(\mathrm{compound}\right)}$$
where MM corresponds to the molar mass. The conversion factors were 1.07 gO_2_/g for glucose, xylose, and acetate; 1.51 gO_2_/g for propionate; 1.82 gO_2_/g for butyrate; 2.04 gO_2_/g for valerate.

#### 2.5.2. Acidification Degree

The acidification degree (AD), which represents the amount of substrate consumed to produce SCOAs, considering the organic matter fed into the batch assays was calculated using Equation (3). These calculations were expressed as percentages.
(3)$$\mathrm{AD}\left(\mathrm{gCOD}/\mathrm{gCOD}\right)=\frac{\mathrm{SCOAproduced}}{\mathrm{CODin}}\times 100$$

#### 2.5.3. Yields and Productivities

Sugar yield was calculated by dividing the total amount of extracted monomeric sugars in the pretreatment by the amount of SCGs, multiplied by 100. COD yield was calculated by dividing the total COD extracted in the pretreatment by the COD of the SCG. SCOAs volumetric productivity was calculated by dividing the amount of produced SCOAs in grams of COD by volume and time.

#### 2.5.4. Odd-to-Even Ratio of SCOAs

With further valorization of SCOAs into PHA in mind, the odd-to-even ratio of acids was calculated to evaluate the potential for each monomer production. It was defined as the sum of odd-equivalent carboxylic acids formed (propionic and n-valeric acids) divided by the sum of even-equivalent carboxylic acids formed (acetic and n-butyric acids), according to Equation (4).
(4)$$\mathrm{Odd}\text{-}\mathrm{to}\text{-}\mathrm{Even}\;\mathrm{Ratio}=\frac{\left[\mathrm{Propionic}\right]+[n\text{-}\mathrm{Valeric}]}{\left[\mathrm{Acetic}\right]+[n\text{-}\mathrm{Butyric}]}$$

## 3. Results and Discussion

### 3.1. Pretreatments

#### 3.1.1. Pretreatment Efficiency

The filtrate obtained after each pretreatment was analyzed to assess the monosaccharides concentration and COD, and the respective yields were calculated as presented in Table 1, except for the supercritical CO_2_ extraction pretreatment (SC), where only a solid phase was obtained. The presence of possible microbial inhibitors, such as TPC and MRPs, after the pretreatment, was also assessed.

The highest sugar concentration, 1.95 g/L, was obtained with the acidic hydrolysis at 5% (AH 5%), followed by the AH 10%, 0.91 g/L, while the rest of the conditions tested led to monosaccharides concentrations below 0.05 g/L. The increase in H_2_SO_4_ concentration seemed to be detrimental to the process as it led to a lower sugar yield. However, AH 10% resulted in a higher COD yield and inhibitor formation, indicating that sugar degradation was occurring [35].

The results were in agreement with the literature, since acidic hydrolysis is often reported as the best pretreatment, either for SCG [22,25] or other lignocellulosic biomasses [47,48]. Compared to previous studies using AH, the concentration of extracted sugars in the supernatant was very low. Using similar acid concentrations, Juarez et al. [22] obtained over 33 g/L of total sugars by using acidic hydrolysis at 5% and 95 °C for 180 min, which corresponded to a yield of 21% (g/gSCG), and Go et al. [23] achieved a yield of 34% when applying 4% of H_2_SO_4_ at 95 °C during 120 min. However, in the present work, only monomeric sugars were quantified, and the contribution of other oligosaccharides was not possible to be determined due to the absence of a proper analytical method. Working with a lower sulfuric acid concentration (0.29%) on SCGs previously submitted to microwave pretreatment, López-Linares et al. [35] obtained a sugar concentration of 5.8 g/L at 170 °C for 5 min. Mussatto et al. [25] reported the highest extraction yield of 50% after also using dilute acidic hydrolysis (~1% of H_2_SO_4_) at 163 °C for 45 min. These results indicate that other parameters such as temperature and solid/liquid ratios also have a significant impact on SCG hydrolysis.

Basic hydrolysis (BH) was reported by some authors as a good pretreatment for SCGs due to high lignin degradation. Girotto et al. [17] tested several concentrations of NaOH (2–8%) for 24 h on SCG before submitting it to anaerobic digestion. Degradation yields of cellulose and hemicellulose (22–25%) were reported; however, the presence of sugars or oligosaccharides was not determined [17]. Ballesteros et al. [28] reported a total sugars yield of 2.38% (g/gSCG) after BH with 4 M NaOH at 25 °C, overnight, on previously defatted and lyophilized SCGs. After obtaining a purified and concentrated extract from hot-water treatment of defatted SCGs, Simões et al. [27] used 4 M NaOH for 2 h at room temperature to extract polysaccharides. The conditions led to a yield of total sugars of 63–76% (g/gSCG). Once again, the pretreatment tested was applied after SCGs were previously submitted to other processes to increase the efficiency of the pretreatment. The inclusion of these extra steps increase costs and complicates its implementation, therefore a simpler strategy should be chosen.

The hydrothermal pretreatment (HD) can be considered as a milder version of this pretreatment due to the lower temperature and pressure applied. Still, it led to interesting results, being the best pretreatment in terms of monosaccharides’ extraction after AH and resulted in lower amounts of inhibitory compounds. Using autohydrolysis to extract polysaccharides from SCGs, Ballesteros et al. [31] reported a yield of total sugars of 33.35% and obtained an extract that was mostly composed of polysaccharides of longer chains. Such can explain the higher COD yield found in the present study.

In general, non-hydrolytic pretreatments (MW and US) led to lower amounts of sugars and COD extracted, which suggests that their impact occurred mostly at a structural level without the full breakdown of sugar chains. This explanation agrees with the literature. Okur et al. [49], using scanning electron microscopy (SEM), observed morphological changes in SCGs after ultrasound treatment to extract TPC. During US treatment, the acoustic cavitation creates micro-fissures and microchannels on the matrix of the SCG, increasing the contact surface and facilitating the biological degradation during the AF step [49]. Similarly, SEM images of SCGs submitted to microwave extraction with superheated water showed the thinning of the cell walls and microstructure changes [50]. Accordingly, the low extraction of sugars and COD in the present work using MW and US was expected, considering the mild conditions used.

Supercritical CO_2_ (SC) extraction is often applied to the by-products of the coffee industry for their valorization. Bioactive oil extracts for pharmaceutical, cosmetic, or nutraceutical fields and triglycerides oil for bioprocesses are the most common products obtained [24,34,41]. Moreover, lipid extraction is often proposed as the first step of SCG-based biorefinery [30,51]. In the present work, this technique was also applied; however, in this case, the outcome was the defatted SCG, and no release of sugars, COD, TPC, or MRPs could be detected.

#### 3.1.2. Inhibitor Formation

The results for inhibitor compounds generated by the pretreatments, namely phenolic compounds and MRPs, are also presented in Table 1. These compounds are known to inhibit several bacterial species [43], and, even though an inhibitory concentration for MMC is hard to define due to its heterogeneity, an impact on its performance is to be expected. Regarding furfural and HMF concentrations, in the main furans produced from SCGs, only trace amounts were detected in the extracts. This probably resulted from the mild extraction conditions used in this study and the short reaction times. The short reaction time was chosen based on several reports about prolonged pretreatment times leading to reduced extraction yields and/or higher concentrations of inhibitors, probably due to the degradation of sugars into furans [22,23]. The absence of these compounds is an advantage for the process, since they can act as microbial inhibitors, hindering the subsequent bioprocesses where SCGs will be used as substrates [43].

TPC was higher in the extracts resulting from hydrolytic pretreatments, with a maximum value for the basic hydrolysis of 30.6 mgGAE/gSCG, which corresponds to a concentration of TPC as gallic acid of 1.05 g/L. This result seems to indicate that the BH 5% is a less suitable pretreatment for biological processes since it led to a lower sugar extraction and higher inhibitor formation. The use of this pretreatment as a detoxification step of defatted SCGs for the removal of lignin was proposed by Passadis et al. [30]. Using 0.7 M NaOH for 6 h at 50 °C, the authors reported 79.2% of lignin degradation. Then, the solid residues were used as substrate for enzymatic hydrolysis [30]. Regarding AH, the increase in acid concentration led to an expected increase in TPC of 2.3 mgGAE/gSCG, similar to what was previously reported by López-Linares et al. [35]. Following the trend of the sugars in the present study, TPC was lower than that reported in the literature. Only a few studies focused on pretreatments for sugar extraction-reported TPC values. Both works studied the use of dilute acidic hydrolysis: Mussatto et al. [25] obtained 3.7 g/L (~37 mgGAE/g) of TPC, and Lopéz-Linares et al. [35] reported values between 0.2 and 1.7 g/L (2–17 mgGAE/g). Some works have suggested that phenolic compounds can be more toxic than other inhibitory molecules at lower concentrations since their low molecular weight allows them to penetrate cell membranes and damage internal structures [43]. Consequently, the small differences in the concentration of TPC obtained after each pretreatment could have a significant impact on the following bioprocessing steps.

Since chlorogenic acid and tannins, the main phenolic compounds in SCGs, retain economic interest, several authors have applied pretreatments to SCGs, aiming to separate them [52]. Getachew et al. [40] reported values between 33 and 51 mgGAE/g for pretreatments using microwaves, ultrasonication, and subcritical water. Conde and Mussatto [53], with a mild hydrothermal pretreatment,, extracted 33 mgGAE/g. Using autohydrolysis, Ballesteros et al. [32] reported 40.36 mgGAE/g. The highest value (230 mgGAE/g) was obtained with basic hydrolysis with 4 M NaOH, though this could be a consequence of the preliminary step of SCG defatting and post-extract lyophilization [28]. As expected, the values were higher than what was obtained in this study but opened the possibility of the separation and extraction of phenolics in a sequential valorization scheme by adapting the conditions of the pretreatment.

Finally, the MRP concentrations on the pretreatments were also evaluated. Absorbance at 294 nm translates into the formation of the intermediate compounds of the MRPs and, at 420, allows for determining their intensity (browning) [42]. The values ranged from 0.234 to 0.831 for absorbance at 294 and from 0.135 to 0.710 for browning intensity. Getachew et al. [40] reported small differences in browning intensity between the pretreatments tested (ultrasonication, microwave, and subcritical water extraction), and the same was observed for US and MW. The highest values were observed again for the BH pretreatment, with considerable differences from the rest of the pretreatments. These results are in accordance with the findings of Bravo et al. [46] when studying the influence of pH on the extraction of antioxidant compounds, where the pH increase led to higher browning intensities.

### 3.2. Acidogenic Fermentation

#### 3.2.1. SCOAs’ Production

AF was performed on the product obtained after each of the pretreatments were tested to understand their impact on SCOAs’ production. The evolution of SCOAs over time for each experiment is represented in Figure 1. The production of SCOAs was observed in all assays, showing that an SCG is a suitable substrate for the process, as some authors have previously shown [15]. Furthermore, except for SCG-SC, all assays resulted in higher SCOAs’ production than the control, an untreated SCG, revealing the efficiency of including a pretreatment step to improve AF. In all assays, the formation of SCOAs was observed immediately upon inoculation and increased until the maximum concentration was achieved, followed by a period of stabilization. When some of the assays were prolonged, the concentrations of SCOAs decreased again, probably due to consumption by the microbial community.

The highest SCOA concentration, 1.33 g/L on day 16, was obtained with SCG-AH 5%, a result in line with the good performance in sugar extraction determined for this pretreatment. It represented an increase of 185% compared to the concentration obtained for the control, which made it the most promising of the options tested. Besides breaking down hemicellulose, acidic hydrolysis is also reported as one of the most effective pretreatment options for protein-rich wastes [21], which could be an advantage given that an SCG still has a significant protein concentration (up to 20%) [54]. Increasing the acid concentration was not beneficial to the process, since SCG-AH 10% led to only 0.80 g/L of acids on day 19. This might be caused by the presence of higher concentrations of inhibitory compounds, degradation of sugars, or due to the higher concentrations of sodium salts formed during the medium neutralization, which were found to have a toxic effect in acetogenic bacteria [21,55].

Besides breaking lignocellulosic structures, basic pretreatment was reported to increase the buffering capacity of the feedstock, preventing pH drops and promoting acidogenesis [21]. However, the acidogenic fermentation of SCG-BH 5% underperformed compared to the other pretreatments, with a maximum production of 0.6 g/L of SCOAs on day 9, which represented an increase of 28% related to the non-pretreated SCGs. Similarly, other works with this pretreatment also reported improvements compared with non-pretreated assays, but low concentrations of SCOAs for the substrates used [56,57,58]. This could be due to the high concentrations of phenolic compounds extracted during the pretreatment (as reported in Table 1)—known inhibitors of mixed microbial cultures [59].

In general, the experimental results showed that the less harsh pretreatments (SCG-US, SCG-MW, and SCG-AC) seemed to result in a better substrate for AF. Even though the release of monomeric sugars and COD was lower, so too was the formation of inhibitors that seemed to have a negative impact on the AF process of SCGs, as could be observed for the other pretreatments. Mild conditions seemed to be enough to increase the accessibility and biodegradability of SCGs and resulted in high concentrations of SCOAs. The maximum SCOA concentration for SCG-US and SCG-MW of 1.06 on day 22 and 1.01 g/L on day 14, respectively, corresponded to an increase of 126 and 115% compared to the control. These techniques combined physical and thermal effects, and their use for waste pretreatment was already associated with an SCOA increases of up to 20 and 5 times for the US pretreatment of waste-activated sludge and MW pretreatment of beet pulp, respectively [60,61]. However, considering the results obtained for SCGs and the high energy demand and maintenance costs of these techniques, SCG-AH, at 5%, still stands as the most attractive option.

SCG-HD also led to an interesting concentration of SCOAs, such as 0.96 g/L on day 16, which was twice the value obtained in the control. Hydrothermal pretreatment, traditionally operated at high temperatures, has gained attention in recent years for use at lower temperatures due to its simple operation and good results [62]. Rice straw was submitted to extraction at 90–130 °C for 15 min, after which Xiang et al. [63] reported an increase of 38% in SCOAs’ yield. A similar increase (31%) was observed by Yuan et al. [64] when pretreating corn stover at 50 °C for 24 h. This may be related to the lower degradation rates of sugars and protein found at lower temperatures. Liu et al. [65] optimized the conditions of the hydrothermal pretreatment of food waste to enhance AF by minimizing the impact of MRPs. A response surface methodology optimized conditions to a temperature of 132 °C, a reaction time of 27 min, and a pH of 5.6, which, when applied, enhanced SCOAs’ production by 22%. Their results also showed temperature to be the most influential parameter, and they found that increasing it over 125 °C for long periods helped to speed up the Maillard reactions and degrade the substrate [65].

SCG-SC had a slightly lower performance compared with the control (−15%), 0.4 g/L on day 9, although the pretreatment was expected to improve SCG biodegradability and increase SCOAs’ production, as reported by some authors [66,67]. These results are still interesting considering that the SCGs already went through a previous step of oil extraction, which did not significantly compromise the concentration of acids. The strategy of the sequential steps of valorization allows for the obtention of more products from SCG, ultimately making the process more viable.

Finally, compared with the other works on the AF of SCGs, the results obtained in the present work were lower than those obtained in batch assays by Girotto et al. [14], namely 30 g/L of SCOAs, after pretreating SCGs with NaOH (8%) at room temperature for 24 h. It is unclear if the difference observed in the results was a consequence of the pretreatment applied or from the substrate itself, since no untreated SCG was used as control. Another possible explanation is the source of MMC used in the work of Girotto and colleagues, which was collected from a full-scale upflow anaerobic sludge blanket digester of a brewery and submitted to thermal treatment to inhibit methanogenic Archaea. Given the lignocellulosic nature of brewery waste, the culture could be already adapted to degrade this type of substrate and less susceptible to inhibitors. On the other hand, working with wastewater rich in SCGs, Arroja et al. [15] obtained a maximum SCOA concentration of 1.33 gCOD/L in a moving bed biofilm reactor after optimization of temperature, organic load rate, and retention times. This value was very similar to the one obtained with SCG-AH 5%. It is important to note that, since this work is a preliminary assay, there is still room for improvement, making the obtained results very promising for a future scale-up.

#### 3.2.2. Acidification Degree

The acidification degree (AD) is often used to access the efficiency of the process [21]. The ADs obtained for each pretreatment are represented in Figure 2, considering the day of maximum production of SCOAs.

The values ranged from 10.0% to 33.5% and can be directly related to SCOAs’ production, with the best results observed for SCG-AH 5%. These values were slightly lower than those previously reported for SCG-related wastes, with values between 20 and 55% [15], depending on the conditions. The overall low values of AD can be explained by the fact that a considerable portion of the COD supplied is composed of recalcitrant compounds that are difficult to degrade by the microbial population. Some of the lower ADs (17.4% for SCG-AC and 12.4% for SCG-BH 5%) corresponded to the pretreatments wherein the highest concentrations of inhibitors were detected. The low biodegradability of SCGs was also verified by Arroja et al. [15] through the determination of a biological oxygen demand (BOD)_5_/COD ratio lower than 0.3. If the ADs would be calculated based on monomeric sugars COD and not on total COD, their values would be over 100% for all. It shows, as expected, that oligosaccharides and longer chains of sugars were present and used by the culture. This was also verified by Queirós et al. [13] when working with a byproduct from the pulp and paper industry. While total AD was relatively low (36%), the AD considering only sugars was 154%, indicating that the culture was consuming more than the monomeric sugars present. This difference indicated that the majority of COD fed to the reactor corresponded to compounds not consumed during the residence time applied [13].

Studies about the acidification of SCGs are still scarce, and these values can mainly be compared with other wastes already tested. Similar values of AD were found for waste sludge, such as 21% [68]; for cassava wastewater, 36% [69]; or for palm oil mill effluent, 25% [70]. Higher ADs were generally reported for substrates with higher biodegradability and a lower presence of inhibitors, such as potato solid waste (69%) [71], cheese whey (67%) [72], food waste (47%) [73], or sugarcane molasses (42%) [74].

#### 3.2.3. Productivity

The ideal acidogenic fermentation should be a continuous process, maximizing the degree of acidification and productivity [21]. These parameters are highly influenced by the reactor operating conditions, such as residence times, pH, and C/N ratios [75,76], which were not tested in this preliminary study. The evaluation of the volumetric productivity of the process can give some insight into the rate of the degradation of the SCGs. The values were calculated considering the day on which the maximum concentration of the SCOAs was reached. By analyzing Figure 2, the lowest productivity, 0.044 gCOD/L·h, was determined for the control, and the highest productivity, 0.104 gCOD/L·h, was still obtained by the SCG-AH 5%, representing a 138% increase. For the non-hydrolytic pretreatments, the tendency of productivity followed the SCOAs’ concentration, with SCG-US resulting in higher productivity than SCG-MW, which could be explained by the improvement in the enzymatic activity that hastened the production of the acids reported for ultrasounds [21]. The third highest productivity was obtained by SCG-HD, a value only 15% below what was achieved with SCG-UA. SCG-BH 5% had the fourth best productivity, 0.072 gCOD/L·h, despite the low performance in SCOA concentration. Finally, SCG-SC had higher productivity than the control, which indicated that, even though it did not result in higher SCOA production, the CO_2_ supercritical pretreatment made the SCGs more accessible to the microorganisms and the degradation was faster.

#### 3.2.4. Odd-To-Even Carbon Ratio of SCOA

To evaluate the potential for valorization into the PHA of the acidified stream produced, it is important to access the SCOA profile. The monomeric composition of PHA, hydroxybutyrate (HB) and hydroxyvalerate (HV), depends on the type of SCOAs available, which can limit further industrial applications for these polymers [77]. The current metabolic model for PHA production from SCOAs proposes that even-number carbon sources (i.e., acetate and butyrate) produce mainly HB, while odd-number carbon sources (i.e., propionate and valerate) lead to the formation of HV. Determining the odd-to-even ratio of SCOAs can give insight into the type of polymer that can be produced. Co-polymers with high HV contents have better thermal and mechanical properties for industrial applications, and, as such, high odd-to-even ratios are more desirable [78].

The composition in odd and even acids formed with each pretreated SCG is represented in Figure 3 and was, for most assays, the same as for the control, namely 87% even acids, with variations reaching below 5%. Previous works on the AF of SCGs obtained similar profiles [14,15], indicating that the substrate source plays a significant role in the type of acids produced. Still, some differences could be observed: acid hydrolysis led to the odd-to-even ratio closest to the control (0.16), with 0.15 and 0.12 for SCG-AH 5 and 10%, respectively. SCG-US and SCG-MW had very similar results (0.28 and 0.27), and SCG-HD (0.38) had a slightly higher content of odd acids. SCG-BH 5% had the lowest odd-to-even ratio (0.03), which could be due to the inhibition of certain microbial communities given the high concentration of phenolic compounds. The highest ratio was obtained using SCG-SC (0.66), which was the most interesting result for PHA production [78,79] and could be an indication that the lipid fraction of the SCG leads to the formation of odd acids.

## 4. Conclusions

The application of a pretreatment step proved to be an efficient strategy to enhance the AF of SCGs since almost all pretreatments resulted in higher concentrations of SCOAs. When selecting the type of pretreatment, there is an important balance to consider between unlocking sugar availability and extracting inhibitory compounds that could be harmful in the subsequent fermentation processes. In this work, the best results were obtained with milder pretreatments.

AH 5% proved to be the best pretreatment applied, and although this technology has some drawbacks (high costs of reagents and equipment), it is still a cheaper alternative to US and MW, and the low pH at the end of the pretreatment could be an advantage in avoiding degradation of the substrate during storage.

SC pretreatment also delivered interesting results. The extraction of the lipidic fraction only resulted in a slight decrease in SCOA production, making the sequential valorization of SCGs through this process feasible. Furthermore, the AF of SCGs submitted to this pretreatment resulted in an acid composition with the most interesting odd-to-even carbon ratio of SCOA for PHA production. However, when considering a full-scale operation, this is a technology that demands higher investment in equipment and trained personnel, making it less attractive when compared to AH.

The results of this work pave the road for the conversion of SCGs into PHA following the concept of a circular economy. The use of pretreatments will facilitate the access of microorganisms to the carbon compounds, allowing the establishment of a three-step process.

## Figures and Tables

**Figure 1 bioengineering-09-00362-f001:**
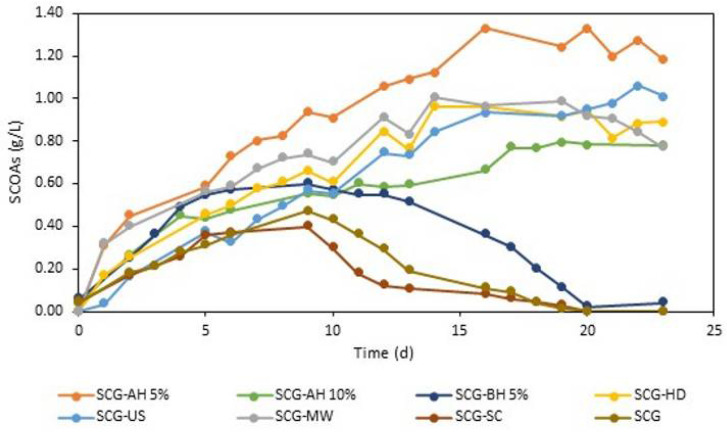
SCOAs’ production over time using the pretreatments tested.

**Figure 2 bioengineering-09-00362-f002:**
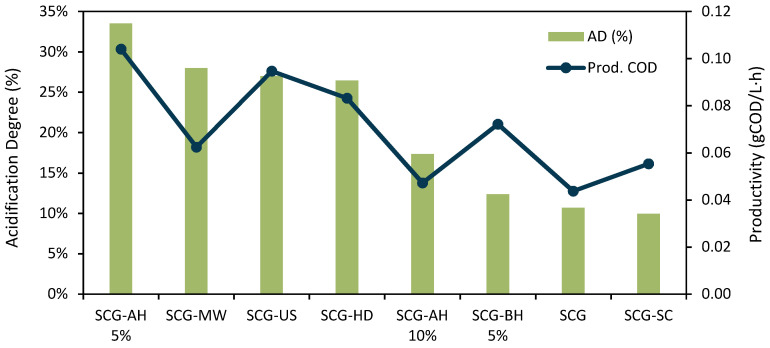
Acidification degrees and productivities for each assay.

**Figure 3 bioengineering-09-00362-f003:**
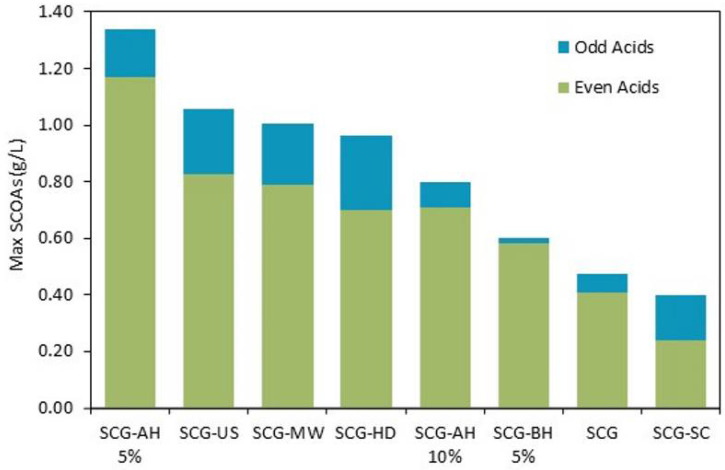
Maximum SCOAs obtained and respective odd and even acid profiles.

**Table 1 bioengineering-09-00362-t001:** Characterization of the filtrate obtained for each pretreatment conducted.

Pretreatment	pH	Sugars (g/L)	%Yield _Sugars_ (gSugar/gSCG)	COD _Extracted_ (gCOD/L)	%Yield _Extracted_ (gCOD/gCOD)	TPC (mgGAE/g)	MRP	
							ABS_294_	ABS_420_
Acidic Hydrolysis 5% (AH 5%)	1.38	1.95	5.20%	4.15	2.25%	21.5	0.438	0.223
Acidic Hydrolysis 10% (AH 10%)	0.50	0.91	2.43%	13.70	8.30%	23.8	0.488	0.240
Basic Hydrolysis 5% (BH)	12.26	0.01	0.03%	11.12	6.74%	30.6	0.831	0.710
Hydrothermal (HD)	6.78	0.05	0.13%	2.95	1.79%	16.8	0.409	0.163
Ultrasounds (US)	7.02	0.01	0.03%	2.38	1.44%	10.1	0.234	0.135
Microwave (MW)	6.94	0.02	0.05%	2.25	1.36%	13.7	0.286	0.142

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
