# Peer review of "Impact of a Pretreatment Step on the Acidogenic Fermentation of Spent Coffee Grounds"

_bioengineering, 2022, doi:10.3390/bioengineering9080362_

Round 1

Reviewer 1 Report

This paper is about the pretreatment of spent coffee grounds to produce short-chain organic acids.

General comments

There are too many citations of the literature in the text. The most important citations should be selected.

The expression “odd-to-even ratio” should include the subject to which it is related.

Materials and methods

In some parts of the text about the used equipment, countries of production have been added, in others, they are not. Please unified. For example, Line 188: There is a lack of the county after “Shimadzu UVmini – 1240”).

Line 107: The “2.2. SCG Pretreatments” should be written in italics.

Results and Discussion

Table 1. The brackets in the word “Sugars” should be deleted.

Figure 2. The legend to the figure should be added.

Conclusions

If it is possible, the conclusion part should include the information on the usage of the pretreatment methods on the full scale.

Author Response

The authors thank Reviewer 1 for the detailed comments that contributed to the correction and a significant improvement of this article. All changes were made directly in the document, highlighted by the "track change" tool.

Comment 1: There are too many citations of the literature in the text. The most important citations should be selected.

To reduce the number of citations, five citations were removed from the introduction. In the other sections, due to the high number of pretreatments tested, the number of papers cited could not be reduced since it would impact the discussion of the results obtained.

Comment 2:  The expression “odd-to-even ratio” should include the subject to which it is related.

The expression “odd-to-even ratio” was clarified in all sections of the document.

Comment 3: In some parts of the text about the used equipment, countries of production have been added, in others, they are not. Please unified. For example, Line 188: There is a lack of the county after “Shimadzu UVmini – 1240”).

The missing information was introduced as suggested.

Comment 4: Line 107: The “2.2. SCG Pretreatments” should be written in italics.

Changes were made to the document as suggested.

Comment 5: Table 1. The brackets in the word “Sugars” should be deleted.

Changes were made to the document as suggested.

Comment 6: Figure 2. The legend to the figure should be added.

Changes were made to the document as suggested.

Comment 7: Conclusions

If it is possible, the conclusion part should include the information on the usage of the pretreatment methods on the full scale.

A comment on full-scale usage of the best pretreatments was added to the conclusion. To fully discuss this, a techno-economic evaluation should be conducted, but the work is still in a preliminary stage.

Reviewer 2 Report

This study details the effect of different pre-treatment steps on the production of short-chain organic acids through the fermentation of spent coffee grounds. The experimental information is sufficiently detailed and a good comparison to the results of similar literature studies is reported. Although the manuscript is well written, some integrations are required to improve the paper before acceptance.

As follows some comments and suggestions:

l.106 Why a temperature of 95 °C was used? Usually, a temperature range between 100 and 105 °C is used.

l. 129-134 A mineral medium was described but it was not mentioned in the experimental setup. It is necessary to describe the quantities and the experimental conditions used for the mineral medium in the fermentation process.

l.118-120 and 147-149 The purification procedure prior to determination of SCOAs, sugars and furans is not clear. According to what I could understand, it should be as follows: 1) filtration with a glass microfiber filter 629 with a pore diameter of 1 μm; 2) filtration with a membrane of 0.2 μm; 3) centrifugation at 8000 rpm. Is it correct? Perhaps it is better to insert a scheme or a figure that describes the procedure.

l.157, 166, 188 A calibration curve was used for the quantification of SCOAs, sugars, furans, COD, TPC but no information was provided. It is necessary to report the calibration curves with some comments.

l.200 (Equation 2). It is necessary to define M.

l.244-245 “…only monomeric sugars were quantified and the contribution of other oligosaccharides was not possible to be determined.” Why?

l. 292 (Inhibitor formation). What is the concentration limit that makes TPC and MRB microbial inhibitors?

l.396 “…high energy demand and maintenance costs of these techniques, SCG-AH 5% still stands as the most attractive option”. This information is important for any industrial scale-up. It is necessary to deepen this issue by reporting, for example, the energy, mass and/or economic balance of the various pre-treatment techniques.

l.437-438, 456-457 (Figure 2). “The ADs obtained for each pretreatment are represented in Figure 2, considering the day of maximum production of SCOAs.” The authors should insert the day of maximum production of SCOAs for each pre-treatment technique in Figure 2.

l.458 Figure 2: it is necessary to add a legend to the figure in order to associate each graph to the respective set of data (is bar chart referring to acidification degree or productivity?)

l.556-557 (ref. #6) The internet link does not work. It is necessary to replace it with a working one or use another reference.

l.657-658 The reference to the DOI link is missing.

l.682-683 The reference to the DOI link is missing.

l.716-717 The reference to the DOI link is missing.

Author Response

The authors thank Reviewer 2 for the detailed comments that contributed to the correction and a significant improvement to this article. All changes were made directly in the document, highlighted by the "track change" tool.

This study details the effect of different pre-treatment steps on the production of short-chain organic acids through the fermentation of spent coffee grounds. The experimental information is sufficiently detailed and a good comparison to the results of similar literature studies is reported. Although the manuscript is well written, some integrations are required to improve the paper before acceptance.

Thank you very much for your appreciation

As follows some comments and suggestions:

Comment 1: l.106 Why a temperature of 95 °C was used? Usually, a temperature range between 100 and 105 °C is used.

The temperature used was 105 °C, the typo was corrected in the text.

Comment 2: l. 129-134 A mineral medium was described but it was not mentioned in the experimental setup. It is necessary to describe the quantities and the experimental conditions used for the mineral medium in the fermentation process.

The description of the mineral medium and experimental procedure was changed to clarify how and when it was used.

Comment 3: l.118-120 and 147-149 The purification procedure prior to determination of SCOAs, sugars and furans is not clear. According to what I could understand, it should be as follows: 1) filtration with a glass microfiber filter 629 with a pore diameter of 1 μm; 2) filtration with a membrane of 0.2 μm; 3) centrifugation at 8000 rpm. Is it correct? Perhaps it is better to insert a scheme or a figure that describes the procedure.

The description of 118-120 corresponds to the filtration conducted to obtain the liquid phase after the application of pretreatment before it was used on AF. The description of 147-149 is the procedure to prepare the samples for HPLC, the filtration occurs during a centrifugation process in microcentrifuge tube filters. Nevertheless, the text was altered to clarify the protocol used.

Comment 4: l.157, 166, 188 A calibration curve was used for the quantification of SCOAs, sugars, furans, COD, TPC but no information was provided. It is necessary to report the calibration curves with some comments.

Usually, calibration curves, done as part of a routine analytical method, are not reported. They were newly done in each batch of samples injected to HPLC to ensure the most accurate results. However, comments on the linearity of calibration curves and the frequency of the injection of standards were added to the text for all the methods that require them.

Comment 5: l.200 (Equation 2). It is necessary to define M.

M was molar mass, but more correctly it was changed to MM and its definition was added to the text.

Comment 6: l.244-245 “…only monomeric sugars were quantified and the contribution of other oligosaccharides was not possible to be determined.” Why?

The contribution of oligosaccharides was not quantified because, unfortunately, it was not possible to have access to this technique during this work. That information was added to the text.

Comment 7: l. 292 (Inhibitor formation). What is the concentration limit that makes TPC and MRB microbial inhibitors?

The concentration limit that makes TPC and MRB microbial inhibitors are highly dependent on the microbial species, and as such is particularly hard to define for mixed cultures of unknown composition. But to make it clear for the reader and enrich the discussion, a comment on the inhibitory effect of these compounds was added to the manuscript.

Comment 8: l.396 “…high energy demand and maintenance costs of these techniques, SCG-AH 5% still stands as the most attractive option”. This information is important for any industrial scale-up. It is necessary to deepen this issue by reporting, for example, the energy, mass and/or economic balance of the various pre-treatment techniques.

Although this would be a valuable discussion, it is not an objective of the paper since this is still a preliminary study that focuses on the preparation of the raw material to be used in the first step of a three-step of PHAs production process. The techno-economic evaluation of the process will depend on how the acidification will be conducted, which was expected to be a continuous system but that will be the subject of a future paper. Still, a comment on the scale-up potential of the considered most promising pretreatments was added to the conclusion.

Comment 9: l.437-438, 456-457 (Figure 2). “The ADs obtained for each pretreatment are represented in Figure 2, considering the day of maximum production of SCOAs.” The authors should insert the day of maximum production of SCOAs for each pre-treatment technique in Figure 2.

The day of maximum production for each pretreatment was included in the discussion, as suggested.

Comment 10: l.458 Figure 2: it is necessary to add a legend to the figure in order to associate each graph to the respective set of data (is bar chart referring to acidification degree or productivity?)

The figure was modified to include the missing legend.

Comment 11: l.556-557 (ref. #6) The internet link does not work. It is necessary to replace it with a working one or use another reference.

The link was fixed.

Comment 12: l.657-658 The reference to the DOI link is missing.

The DOI link was added.

Comment 13: l.682-683 The reference to the DOI link is missing.

The DOI link was added.

Comment 14: l.716-717 The reference to the DOI link is missing.

The DOI link was added.

Reviewer 3 Report

Overall, this is an interesting study focusing on pretreatment of SCG followed by AF for production of VFA. The manuscript is readable. Some suggestions are provided for the authors to improve the manuscript. 

1. The standard deviations of all the data in Figures and Tables should be supplemented to increase the scientific rigor.

2. The produced VFA is still mixed with the fermentation broth, how did the authors separate the VFA from the mixture would be a problem. The authors are suggested to supplement some discussion on the aspect to make this study more practical.

3. The fermentation is a batch mode fermentation; the authors are suggested to discuss the potential study by using a continuous or semi-continuous mode.

4. The elemental analyses of the SCG can be supplemented for helping future related research and application. 

5. How to treat the fermentation broth after finishing the AF process, especially in a potential industrial scale? 

Author Response

The authors thank Reviewer 3 for the detailed comments that contributed to the correction and a significant improvement to this article. All changes were made directly in the document, highlighted by the "track change" tool.

Overall, this is an interesting study focusing on pretreatment of SCG followed by AF for production of VFA. The manuscript is readable. Some suggestions are provided for the authors to improve the manuscript.

Thank you very much for your appreciation

  1. The standard deviations of all the data in Figures and Tables should be supplemented to increase the scientific rigor.

Due to the high number of assays, the limited volume of samples and their number to analyze, they were not conducted in duplicate, as such, there was no calculation of the standard deviations.

  1. The produced VFA is still mixed with the fermentation broth, how did the authors separate the VFA from the mixture would be a problem. The authors are suggested to supplement some discussion on the aspect to make this study more practical.

As mentioned in the introduction, short-chain organic acids (SCOA) produced during acidogenic fermentation will be used for the biosynthesis of polyhydroxyalkanoates (PHA). Acidogenic fermentation is the first step of the three-step process. In the second and third steps, the mixtures of SCOA are used to enrich and feed the PHA-producing mixed microbial culture (MMC). For this reason, in our case, we do not need to separate the acids. In the future, we need to control the composition of these mixtures of SCOA, because it will allow tuning the monomeric composition of PHA produced. This control is done by selecting the right operational conditions after the acidogenic reactor is established. For now, we are still testing the best way to prepare the spent coffee grounds for the acidogenic step.

  1. The fermentation is a batch mode fermentation; the authors are suggested to discuss the potential study by using a continuous or semi-continuous mode.

This work aimed to be a screening of potential pretreatments for AF of SCG, which is planned to be applied to a bioreactor working in continuous mode. Because of the high complexity of the substrate and to fully understand the impact on the MMC it was important to conduct the AF assays with the pretreated spent coffee grounds. The next step is to confirm the results of the preliminary assays for the best pretreatments at the bioreactor scale.

  1. The elemental analyses of the SCG can be supplemented for helping future related research and application.

The elemental analysis of the SCG is not essential for the project at this moment and it is also well documented in the literature. As such, we did not include it. As general reference two examples are cited in the paper: references 3 (Mata, Martins, & Caetano, 2018) and 33 (Passos & Coimbra, 2013).

  1. How to treat the fermentation broth after finishing the AF process, especially in a potential industrial scale?

As mentioned before, this is a preliminary study that will evolve into the first step of a three-step production process, working in a bioreactor on a continuous mode. The fermentation broth resulting from AF is to be used as feedstock for the other two steps of the process, after centrifugation or filtration to remove the acidogenic microorganisms. No treatment is anticipated to be needed, which denotes the suitability of this type of valorization for this waste.

Reviewer 4 Report

This work require extensive review. It looks just laboratory report

This manuscript studied Acidogenic fermentation of Spent coffee grounds by the addition of a various pre-treatment mechanisms including acidic hydrolysis, basic hydrolysis, hydrothermal, microwave, ultrasounds, and supercritical CO2. Performances of these pre-treatments were ultimately evaluated in terms of their sugar and inhibitors release. The authors collected limited experimental data and few analyses, The scope is limited and its relevance to the scientific community is minimal. The manuscript also suffers from various linguistic concerns. I, therefore, advise significant review before publication. More comments.

- Hidrothermal ?? check spelling throughout the manuscript.

- Is spent coffee really a major environmental hazard? Why do we need to invest such substantial effort to pre-treat and then biodegrade coffee? (is it because its environmental hazard or there is some sort of valuable resource to recover) please explain the premise/hypothesis behind employing coffee residue as a substate. How much short-chain organic acids and aliphatic monocarboxylic acids can a coffee potentially produce (include from literature). Include also its methane yield.

- What is the conclusion of this work? is both AH and SC the best pre-treatment ?

- The scope is too narrow. It would be interesting to consider dominant microbial communities governing the SCOAs production   

- How do you prove that the pre-treatment experiment was still in the acidogenic stage at 25 days. The methane yield must be measured (at least after 20 days) to prove that the digestion was purely at acidogenesis. Fig 1 is not properly explained.

- Explain how this work pertain to the circular economy

Author Response

The authors thank Reviewer 4 for the detailed comments that contributed to the correction and a significant improvement to this article. All changes were made directly in the document, highlighted by the "track change" tool.

This work require extensive review. It looks just laboratory report

This work aimed to be a screening of potential pretreatments for AF of SCG, which will represent the first step of a three-step production process to produce polyhydroxyalkanoates. The choice of the pretreatments applied was based on the most promising literature results and, besides analyzing the results of the pretreatment in terms of sugars and inhibitors, the pretreated SCGs were then submitted to AF to fully understand the impact of pretreatment of such a complex substrate on the microbial conversion into SCOA. Throughout the manuscript, the results obtained were thoroughly discussed and compared with those obtained by other authors. We do consider the work has a significant quality and can be of interest to Bioengineering journal audience, mainly after the reviewer’s comments/improvements.  

This manuscript studied Acidogenic fermentation of Spent coffee grounds by the addition of a various pre-treatment mechanisms including acidic hydrolysis, basic hydrolysis, hydrothermal, microwave, ultrasounds, and supercritical CO2. Performances of these pre-treatments were ultimately evaluated in terms of their sugar and inhibitors release. The authors collected limited experimental data and few analyses, The scope is limited and its relevance to the scientific community is minimal. The manuscript also suffers from various linguistic concerns. I, therefore, advise significant review before publication.

Given the current transition into green chemistry and circular economy, the interest in residues valorization, particularly for PHA production, has grown significantly in the last years (De Donno Novelli, Moreno Sayavedra, & Rene, 2021). This approach allows not only an end of life for residues but also reduces PHA costs of production. To achieve this, a three-step production process is used, encompassing (1) an acidification reactor to produce short-chain organic acids (SCOA) the preferred precursors of MMC to produce PHA; (2) an MMC selection reactor that, using the SCOAs-enriched stream obtained from acidification, imposes the selective pressure to enhance the survival of microorganisms with PHA storage ability; and, finally, (3) a fed-batch reactor where the selected MMC accumulated PHA at maximum capacity. In this work, we focused on a screening of pretreatments to be later applied to the first step, as unlocking the sugar potential of residues is one of the bottlenecks of the process. Therefore, we believe this is a relevant scientific contribution, not only to the acidogenic fermentation field and valorization of an industrial residue but also as an insight for everyone working with pretreatments to be applied to biological processes.

More comments.

- Hidrothermal ?? check spelling throughout the manuscript.

The typo was corrected in the abstract.

- Is spent coffee really a major environmental hazard? Why do we need to invest such substantial effort to pre-treat and then biodegrade coffee? (is it because its environmental hazard or there is some sort of valuable resource to recover) please explain the premise/hypothesis behind employing coffee residue as a substate. How much short-chain organic acids and aliphatic monocarboxylic acids can a coffee potentially produce (include from literature). Include also its methane yield.

Spent coffee grounds are an industrial residue with a high environmental impact. As stated in the introduction 9.9 megatons of coffee beans were produced in 2019/2020, for every ton of coffee 650 kg of residues are generated. Although some waste valorization approaches have been developed, they only correspond to a small percentage of the waste, and the majority end up in landfills. SCG is characterized by high organic content, in the form of insoluble polysaccharides bound to the material, fatty and amino acids, polyphenols, and minerals, that can be exploited as a source of value-added products (Kourmentza et al., 2017; Mata et al., 2018). Furthermore, some of these compounds like tannins, polyphenol, and caffeine are highly toxic and if disposed of in the environment can cause significant contamination (Rajesh Banu et al., 2020). There are many works on SCG valorization, as mentioned in several reviews on the subject, and several biorefineries on the topic were proposed. Many of these are mentioned in the introduction but were not further explored because it wasn´t the focus of this work.

Regarding its potential for SCOA production, only a few studies focused on this, and they are all described in the introduction/discussion of the manuscript. For methane production, the field is a bit more developed and some of those works were mentioned in the introduction, but production metrics weren’t discussed as methane production was not the product we aimed to obtain. The yields of methane vary on whether its mono-digestion (336 ± 7 mL CH4/g VS reported by (Atelge et al., 2021) – Reference 16 on the manuscript) or co-digestion, a frequently used strategy as SCG reported to improve biogas production (1.4 Lbiogas/Lreactor/d) (Orfanoudaki et al., 2020). Both of these works consider the conversion of organic matter into methane as a final product which was not the aim of this work. Here the anaerobic digestion process will mostly remain in the acidogenic/acetogenic step, before methanogenesis.

- What is the conclusion of this work? is both AH and SC the best pre-treatment ?

Both, AH and SC, showed promising results. However, to make the conclusion clear, the discussion was further expanded on the potential impact of scale-up using both strategies, which accentuates the benefits of AF.

- The scope is too narrow. It would be interesting to consider dominant microbial communities governing the SCOAs production  

As mentioned above, this work should serve as a screening of pretreatments, for later be applied to a bioreactor working in continuous mode. In that bioreactor, the evolution of the dominant microbial communities over the operational time will be analyzed.

- How do you prove that the pre-treatment experiment was still in the acidogenic stage at 25 days. The methane yield must be measured (at least after 20 days) to prove that the digestion was purely at acidogenesis. Fig 1 is not properly explained.

In this study we wanted to understand which pretreatment resulted in the highest SCOA concentration and productivity, the aim is to apply the best pretreatment to a continuous bioreactor, that should operate at a retention time under 15 days. In that bioreactor, methane production would be evaluated but using such a low retention time we expect methanogenesis to be residual. Furthermore, in this work, an aerobic inoculum was used instead of the typical anaerobic one, as this was reported as a good strategy to prevent methanogenesis (Queirós, Sousa, Pereira, & Serafim, 2017).

- Explain how this work pertain to the circular economy

The environmental impact and importance of the valorization of SCG were already mentioned in the answers above. When applied to waste management circular bioeconomy focus on the transformation of the waste into multiple products, fitting a biorefinery concept. In this work, we show that another bioproduct can be produced from waste (SCOA) and propose it to be further valorized into bioplastics. But we also evaluate the possibility of using a pretreatment, supercritical CO2 extraction, that removes even more added value from SCG without a significant impact in the next stage of the process, as discussed in 3.2.1. Such a result can be helpful to tailor future SCG biorefineries.

Atelge, M. R., Atabani, A. E., Abut, S., Kaya, M., Eskicioglu, C., Semaan, G., Kumar, G. (2021). Anaerobic co-digestion of oil-extracted spent coffee grounds with various wastes: Experimental and kinetic modeling studies. Bioresource Technology, 322(November 2020), 124470. https://doi.org/10.1016/j.biortech.2020.124470

De Donno Novelli, L., Moreno Sayavedra, S., & Rene, E. R. (2021). Polyhydroxyalkanoate (PHA) production via resource recovery from industrial waste streams: A review of techniques and perspectives. Bioresource Technology, 331(December 2020), 124985. https://doi.org/10.1016/j.biortech.2021.124985

Kourmentza, C., Plácido, J., Venetsaneas, N., Burniol-Figols, A., Varrone, C., Gavala, H. N., & Reis, M. A. M. (2017). Recent Advances and Challenges towards Sustainable Polyhydroxyalkanoate (PHA) Production. Bioengineering, 4(2), 55. https://doi.org/10.3390/bioengineering4020055

Mata, T. M., Martins, A. A., & Caetano, N. S. (2018). Bio-refinery approach for spent coffee grounds valorization. Bioresource Technology, 247(September 2017), 1077–1084. https://doi.org/10.1016/j.biortech.2017.09.106

Orfanoudaki, A., Makridakis, G., Maragkaki, A., Fountoulakis, M. S., Kallithrakas-Kontos, N. G., & Manios, T. (2020). Anaerobic Co-digestion of Pig Manure and Spent Coffee Grounds for Enhanced Biogas Production. Waste and Biomass Valorization, 11(9), 4613–4620. https://doi.org/10.1007/s12649-019-00796-6

Passos, C. P., & Coimbra, M. A. (2013). Microwave superheated water extraction of polysaccharides from spent coffee grounds. Carbohydrate Polymers, 94(1), 626–633. https://doi.org/10.1016/j.carbpol.2013.01.088

Queirós, D., Sousa, R., Pereira, S. R., & Serafim, L. S. (2017). Valorization of a Pulp Industry By-Product through the Production of Short-Chain Organic Acids. Fermentation, 3(20), 1–11. https://doi.org/10.3390/fermentation3020020

Rajesh Banu, J., Kavitha, S., Yukesh Kannah, R., Dinesh Kumar, M., Preethi, Atabani, A. E., & Kumar, G. (2020). Biorefinery of spent coffee grounds waste: Viable pathway towards circular bioeconomy. Bioresource Technology, 302(November 2019). https://doi.org/10.1016/j.biortech.2020.122821

Round 2

Reviewer 2 Report

The manuscript can be accepted in present form.

Author Response

Thank you very much for the time spent reviewing the paper.

Reviewer 3 Report

There is no standard deviations for the data, which means this research is not very scientific. Hence, the authors are suggested to mention that there was no calculation of the standard deviations and explain the reasons in the mainboday. 

Author Response

The text was modified accordingly to the reviewer's comment.

Reviewer 4 Report

Accept

Author Response

(The authors gave the same response as above.)
